# Treatment outcomes of Nigerian patients with tuberculosis: A retrospective 25-year review in a regional medical center

**Michael A. Alao**[1,2]*, **Stacene R. Maroushek**[3,4], **Yiong Huak Chan**[5], **Adanze O. Asinobi**[6], **Tina M. Slusher**[1,2,3,4]*, **Daniel A. Gbadero**[1,2]

1 Department of Paediatrics, Bowen University Teaching Hospital, Ogbomoso, Oyo State, Nigeria, 2 Bowen University College of Medicine Iwo, Iwo, Osun State, Nigeria, 3 Hennepin Healthcare, Minneapolis, Minnesota, United States of America, 4 University of Minnesota, Minneapolis, Minnesota, United States of America, 5 Yong Loo Lin School of Medicine, National University of Singapore, Singapore, Singapore, 6 University College Hospital Ibadan, Ibadan, Oyo State, Nigeria

* mikevikefountains@gmail.com (MAA); tslusher@umn.edu (TMS)

## Abstract

### Introduction

Tuberculosis (TB) remains a global health challenge and leading infectious killer worldwide. The need for continuous evaluation of TB treatment outcomes becomes more imperative in the midst of a global economic meltdown substantially impacting resource-limited-settings.

### Methods

This study retrospectively reviewed 25-years of treatment outcomes in 3,384 patients who were managed for TB at a tertiary hospital in Nigeria. Confirmed TB cases were given directly observed therapy of a short-course treatment regimen and monitored for clinical response.

### Results

Out of 1,146,560 patients screened, there were 24,330 (2.1%) presumptive and 3,384 (13.9%) confirmed TB cases. The patients' mean age was 35.8 years (0.33–101 years). There were 1,902 (56.2%) male, 332(9.8%) pediatric, and 2,878 (85%) pulmonary TB cases.

The annual mean measured treatment outcomes were as follows: adherence, 91.4(±5.8) %; successful outcome, 75.3(±8.8) % potentially unsatisfactory outcome, 14.8(±7.2) %; and mortality 10.0(±3.6) %.

Female, extra-pulmonary TB (EPTB), newly diagnosed, and relapsed patients compliant with treatment had successful outcomes. Adulthood and HIV infection were mortality risk factors.

### Conclusion

The mean annual successful treatment outcome is 75.3(±8.8) %. Female, pediatric, EPTB, new, and relapsed patients were predisposed to successful treatment outcomes. Lessons learned will guide future program modifications.

**Data Availability Statement:** All data is fully available without restrictions. The data underlying the results presented are available by contacting

Dr. Michael Alao at michael.alao@bowen.edu.ng or the Director of Clinical Services at Bowen University Teaching Hospital Dr. Adewumi, O. Durodola at adewumi.durodola@bowen.edu.ng. Data will be made available to all interested researchers upon request including all supporting data.

**Funding:** The author(s) received no specific funding for this work.

**Competing interests:** The authors have declared that no competing interests exist.

## Introduction

Despite the general improvement in global health standards in the second decade of the 21st Century, the prevalence and mortality of tuberculosis (TB) in low-and middle-income countries (LMICs) remain high [1]. Since 1993, when the World Health Organization (WHO) declared tuberculosis (TB), a disease of increasing global concern, millions of people are still being infected and die annually as a result of TB [1,2]. Multi-drug resistant *Mycobacterium tuberculosis* (MTB) infection is also on the rise, making TB infection of more concern [1,2]. The factors contributing to the persistence of high TB with poor outcomes in LMICs include a high level of extreme poverty, low-quality health care, inadequate nutrition, and overcrowding [1,3].

The third sustainable development goal (SDG3) target is to end the TB epidemic by 2030. In order to achieve this goal, LMIC TB programs must have comprehensive and ongoing reviews of successes and failures in order to guide systematic improvements that will aid in achieving the WHO set target of 85% treatment success. In the past three decades, studies reported varying TB treatment outcomes ranging from 34.0% to 85.45% in the LMICs [1,4]. However, comprehensive data of TB treatment in LMICs are scarce, often of poor quality, or of a limited time span, making these data sub-optimal for informed decision-making.

The present study evaluated data collected over 25 years of a continuing TB treatment program and focuses on the outcomes and lessons that can be learned as the world strives towards achieving the SDG3 targets.

## Materials and methods

This is a retrospective review of the TB treatment program at Bowen University Teaching Hospital (BUTH) formally called Baptist Medical Centre Ogbomoso (BMCO), Nigeria from 1992 to 2017. In the early 1990s, using the National Tuberculosis Leprosy Control Program (NTBLCP) protocol, the Damien Foundation (Netherlands) in collaboration with Oyo State chose BMCO, a regional referral center, as one of the program's initial healthcare facilities. Together BMCO/BUTH, the Damien Foundation and the Oyo State government provided free medications, education, diagnostics, counseling, contact tracing and ongoing updates and reviews. All patients were fully evaluated and had all or some of the following laboratory investigations performed depending on their presenting features: white blood cell count, erythrocyte sedimentation rate, tuberculin skin testing (TST), chest and spinal X-rays, sputum microscopy for acid-fast bacilli (AFB) stain and culture, fine needle aspirate for cytology, histology, and GeneXpert from 2015. Since 2006 all TB patients were screened for HIV1 and HIV2 by parallel testing using Determine® Kits (Alere Medical, Chiba-ken Japan), and Uni-GoldTM HIV (Trinity Biotech, Wicklow, Ireland), and tie-break with STAT-PAK® (Chembio Diagnostic System, Medford NY).

A patient was confirmed to have active TB if any of the following were found.

a. AFB positive sputum microscopy.

b. TST induration >15 mm and/or ulceration or TST of 10mm plus background history suggestive of TB.

c. AFB positive spinal cerebrospinal fluid (CSF) on microscopy.

d. Spinal gibbus with characteristic x-ray.

e. Fine needle aspiration cytology showing granulomatous giant Langhans cells.

f. Positive histopathological tissue findings of chronic granulomatous inflammation with giant cell and caseous necrosis.

g. Positive GeneXpert test on tissue/body fluids.

A definitive diagnosis of TB in children was difficult as most children cannot produce sputum. The NTBLCP workers manual 5 and most recently the Childhood Tuberculosis Desk Guide for diagnosis and management of childhood tuberculosis was used, an adaptation from the NTBLCP workers manual. This entails a composite clinical and laboratory reference standard5 of history and exam plus one or more of the following. Chest X-ray, tuberculin skin test, gastric aspirate, non-response to routine antibiotics in a child with history and exam consistent with TB with subsequent response to TB drugs.

All patients diagnosed with TB received directly observed therapy (DOT) daily in the TB Unit where they received individual education about their disease, treatment course and were observed for complications from the disease or treatment. Home visits were carried out prior to discharge with contact tracing and locating patients who defaulted. Patients were followed up by physicians at two weeks; one, two and five months and as needed unless "lost to follow-up" or transferred to another center.

## Treatment protocol and regimen

Modifications of the treatment protocol did occur over the 25 years in order to follow WHO protocol changes adopted by the NTBLCP. Treatment was divided into categories I and II. Category I was for newly-diagnosed pulmonary TB while category II was for extra-pulmonary TB, treatment failure, or relapse. Between 1992 and 2000, children (<18years) received intramuscular streptomycin instead of oral ethambutol during the intensive/initial 2-month treatment phase. Drug combinations evolved over the years to the current combination of rifampicin, isoniazid, pyrazinamide, and ethambutol (RIPE) for drug-sensitive MTB.

Adherence was defined as patients taking ≥80% of the prescribed anti-TB via the DOT intervention or treatment completed. The study used a daily dosing treatment regimen and calculated adherence for the entire treatment course (no separation was made between the intensive and the continuation phase).

In addition to DOT, patients were encouraged to eat a balanced diet, sleep alone in well-ventilated rooms during the intensive treatment phase and ensure that their sputa were appropriately disposed of to prevent spread of TB to others.

## Definition of treatment outcomes

Treatment outcomes as defined by WHO guidelines and the International Union Against Tuberculosis and Lung Diseases (IUATLD) follow [3,5].

a. Cure: Sputum smears negative on two occasions, one of which must be at treatment end.

b. Treatment completed: Patient completed treatment but last smear unavailable or extrapulmonary TB.

c. Successful treatment is a or b.

d. Relapse: When a patient who had been declared cured later has a positive smear.

e. Treatment failure: Patient remains smear positive ≥5-months after beginning treatment.

f. Default: Patient had ≥1-month of treatment with >2month interruption.

g. Death: Mortality from any cause during the course of receiving treatment.

h.  Transfer out: Patient referred out of catchment area and outcome of treatment unknown.

i.  Potentially unsatisfactory treatment outcome is a combination of treatment default, transferred out and treatment failure [3].

## Data analysis

Statistical analyses were performed using IBM's Statistical Package for Social Sciences (SPSS)$^{TM}$, Version 23.0 for Windows with statistical significance set at $p < 0.05$. Descriptive statistics for categorical variables were presented as number (%). The significant variables of sociodemographic characteristics of age (adult or children), gender (male or female) and clinical variables of categories of diseases (pulmonary, extrapulmonary disease, pre-treatment status, and HIV status) on bivariate analysis were subjected to binary logistic regression to adjust for co-variance to determine the predictors of successful treatment outcomes, treatment adherence potentially unsatisfactory outcome and death. Interaction effects were built from the statistical perspective (only significant independent variables from the main effects multivariate analyses were interacted). The trend in incidence was assessed for significance using a linear regression model.

## Ethical approval

This retrospective review was approved by the Bowen University Teaching Hospital Human Research Ethics Committee (Approval No. BUTH/REC-029). As a result of the retrospective nature of the study design, informed consent was waived by the ethics committee.

## Results

Of the 1,146,560 patients seen in the hospitals' clinics during the study period, 24,330 presumptive cases were investigated for TB and 3,443 (14.2%) cases were confirmed following WHO TB diagnosis guidelines (Fig 1). Fifty-nine (1.7%) patients were excluded because of insufficient data. There were no significant differences in demographic and clinical characteristics between study subjects and those excluded.

### Sociodemographic and clinical characteristics of study population

The median age of patients was 35.7 (range 4 months to 101 years) with 52.5% in the 20–39 years (Fig 2) range and 1,904(56.2%) giving a male:female ratio of 1.3:1. Children accounted for one-tenth 332 (9.8%) of TB cases. The female gender predominate the TB cases in childhood. Fewer children 141 (69.5%) were sputum positive compared with the adults 2020 (75.5%). Additionally, the most common extrapulmonary sites for TB differed between children and adults. New cases constituted 90.3% (3,059). The second largest category were patients who had previously failed TB treatment (132; 3.9%). Other clinical characteristics are shown in Table 1 and Fig 3.

### Annual TB incidence, HIV infection and multi-drug resistance rate

Fig 3 shows the trend in incidence of TB over the 25 years under review. The annual number of newly diagnosed tuberculosis cases peaked in 1993 and 1998. The time trend incidences were 401.3 and 607.7 per 100,000 population respectively as estimated from hospital records (S1 Appendix). The incidence however declined after 2005 with a record of 175.6 cases per 100,000 population in 2014. The incidence in TB among the pediatric population declined by 0.003% (95% CI 0.001% to 0.005%) per year, $p = 0.012$. A decline of 0.010% (95% CI 0.004% to 0.016%) per year, $p = 0.002$ was also noted in adults. The mean decline in the entire population

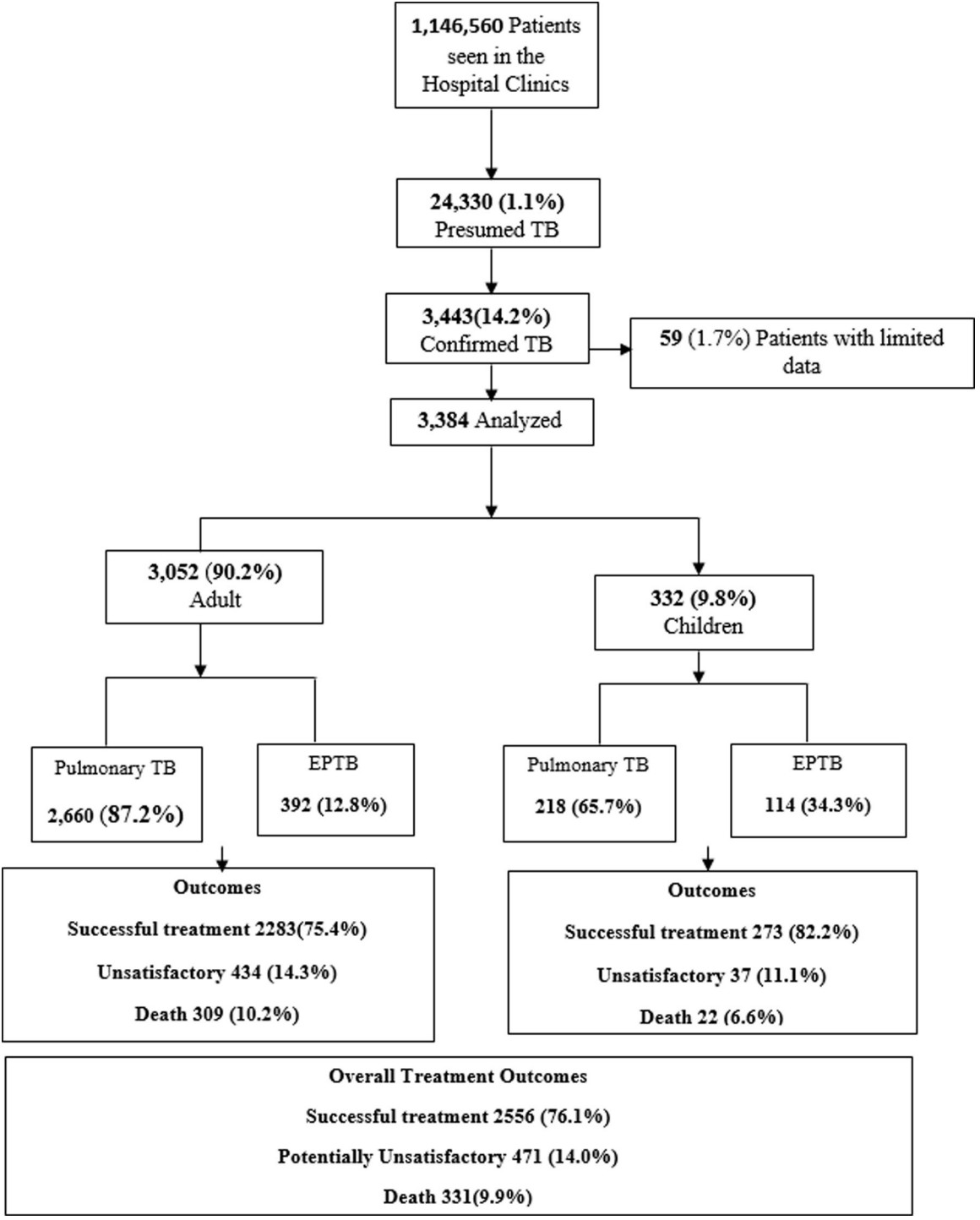

**Fig 1. The flow diagram of program enrolment and treatment outcome of TB cases.** The flow diagram of program enrolment and treatment outcome of TB cases. The final outcome data excludes 26 missing data points (TB:tuberculosis; EPTB: Extrapulmonary tuberculosis).

with TB was 0.007% (95% CI 0.002% to 0.011%) per year, *p* = 0.005. Of the 3,467/24,330 (14.2%) screened for HIV, 1,243 were positive giving a seroprevalence rate of 36.70%.

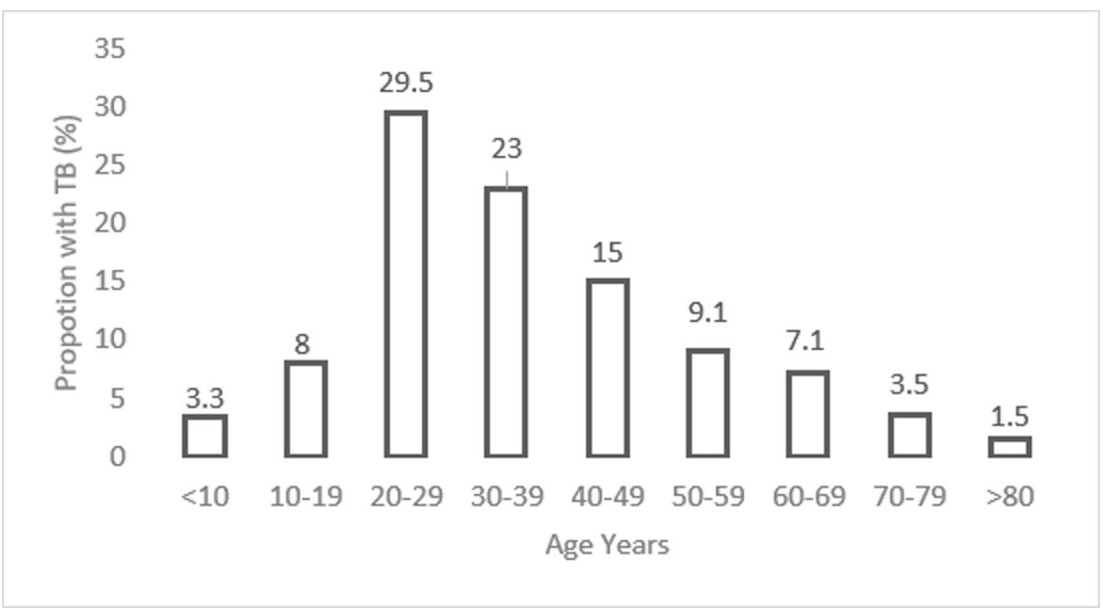

**Fig 2. Age distribution of TB patients.**

The proportion of patients screened for HIV infection over the study period is shown in Table 2. Of the 3,384 diagnosed with TB, only 1,004 were tested for HIV infection out of which 152 tested positive giving a seroprevalence rate of 15.0%. The prevalence shows unstable pattern (sine wave pattern) with peaked in 2009.

## TB treatment outcomes

The mean annual adherence rates over the 25-year study period was 91.4 (±5.8) %. This outcome was fairly constant except for declines in 1994 (78.5%) and 2013 (74.2%) (Fig 4). The mean successful annual treatment outcome was 75.3(±8.8) %, varied between 91.8% in 1992 and 77.8% in 2017. The nadir of successful outcomes occurred in 1994 (58.9%). The mean annual death was 10.0(±3.6) %. Annual mean unsatisfactory treatment outcome was 14.8 (±7.4) % being at all-time low at the outset of the program (1992) but peaked in 1994 (41.1%) (Fig 4).

## Factors associated with adherence

Factors with statistically higher adherence rates were females (90.9% vs 87.4%, aOR = 1.4, 95% CI 1.1–1.7, $p$ = 0.005), EPTB (93.7% vs 88.1%, aOR = 1.8, 95% CI 1.3–2.7, $p$ = 0.002), newly diagnosed TB (94.2%, aOR = 3.1, 95% CI 1.4–6.7, p = 0.005) and relapsed TB (95.1%, aOR = 2.8, 95% CI 1.1–7.5, $p$ = 0.037) subjects. Children and those with positive HIV also displayed higher adherence rates (Table 3). About 301 (10.0%) of the patients with treatment adherent could not submit sputum sample at the end of their treatment course.

## Factors associated with successful treatment outcome

Factors associated with successful treatment outcome were females (80.4% vs 76.1%, aOR = 1.3, 95% CI 1.1–1.5, $p$ = 0.006), EPTB (83.0% vs 77.1%, aOR = 1.3, 95% CI 1.01–1.7, $p$ = 0.036), newly diagnosed TB (79.1%, aOR = 2.7, 95% CI 1.3–5.4, $p$ = 0.005) and relapsed TB (79.4%,

**Table 1. Sociodemographic and clinical characteristics of study population.**

| Variable | Adults n (%) | Children n (%) | Population n (%) |
|---|---|---|---|
| Age in distribution | | | |
| | 3052 (90.2) | 332 (9.8) | 3384 (100.0) |
| Gender | | | |
| FEMALE | 1265 (41.8) | 201 (60.4) | 1466(43.6) |
| MALE. | 1761 (58.2) | 132 (39.6) | 1893(56.4) |
| HIV Status | | | |
| HIV positive | 140 (4.6) | 12 (3.6) | 152 (4.5) |
| HIV negative | 782 (25.8) | 71 (21.3) | 852 (25.2) |
| HIV Not tested/status unknown/ | 2104 (69.5) | 249 (75.0) | 2380 (70.3) |
| Type of Tuberculosis | | | |
| PTB | | | |
| Sputum Positive | 2020 (75.5) | 141 (69.5) | 2161 (75.1) |
| Sputum Negative | 655 (24.5) | 62 (30.5) | 717 (24.9) |
| EPTB | 392(12.8) | 114 (34.3) | 506 (100.0) |
| Pleural Effusion | 4 (3.6) | 53 (13.5) | 57 (11.3) |
| TB Arthritis | 5 (4.4) | 12 (3.1) | 17 (3.3) |
| TB Spine | 50 (43.8) | 234 (59.7) | 284 (56.1) |
| TB Abdomen | 18 (15.8) | 27 (6.9) | 45 (8.9) |
| Miliary | 3 (2.6) | 11 (2.8) | 14 (2.8) |
| Adenitis | 31 (27.2) | 50 (12.7) | 81 (16.0) |
| Endometriosis | 0 (0.0) | 1 (0.3) | 1 (0.2) |
| Meningitis | 2 (1.8) | 1 (0.3) | 3 (0.6) |
| TB Pericarditis | 1 (0.8) | 3 (0.7) | 4 (0.8) |
| Pre-Treatment status | | | |
| New | 2716 (89.8) | 317 (95.2) | 3033 (90.3) |
| Relapse | 96 (3.2) | 6 (1.8) | 102 (3.0) |
| Transfer | 6 (0.2) | 0 (0.0) | 6 (0.2) |
| Default | 30(1.0) | 3(0.9) | 33 (1.0) |
| Failure | 127(4.2) | 5 (1.5) | 132 (3.9) |
| Others | 51 (1.7) | 2 (0.6) | 53 (1.6) |

The variables of gender 25 (0.7%), HIV status 26(0.8%), and pre-treatment status 21 (0.6%) have the respective missing data.

aOR = 2.9, 95% CI 1.3–6.8, $p$ = 0.012) compared with defaulters. Trend was close to significance for children and negative HIV patients (Table 4).

## Factors associated with death/unsatisfactory outcomes

Adulthood (9.9% vs 5.4%, aOR = 1.9, 95% CI 1.2–3.1, $p$ = 0.011) and HIV+ (16.4% vs 8.7%, aOR = 2.1, 95% CI 1.3–3.5, $p$ = 0.003) were risk predictors for death (S2 Appendix). The male gender, pulmonary tuberculosis, relapse and smear positivity at two and five months of treatment were significantly associated with potentially unsatisfactory outcome (S3 Appendix).

## Discussion

The sociodemographic characteristics of TB cases in the current review shows a male preponderance with a gradual rise in incidence from pubertal years with peak at age 20–39 years similar to reports in literature [6–8]. The second-to-third decade of life has been associated with

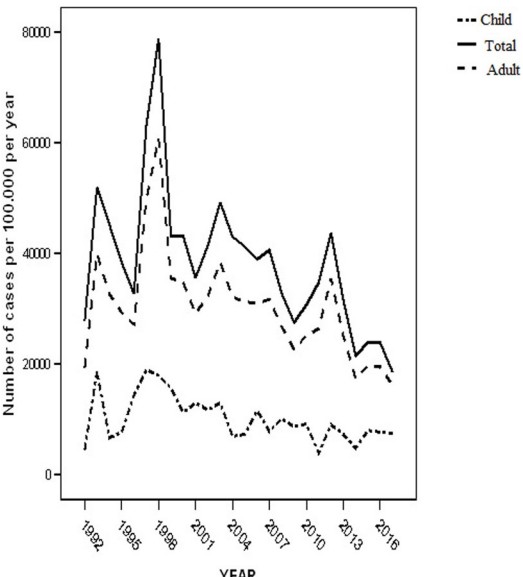

**Fig 3. Annual incidence of TB.** The figure shows the annual incidence of tuberculosis for children and adult over the 25-year study period.

high-risk behaviours such as drug abuse, smoking, alcoholism, socioeconomic struggles (poverty, poor housing) which are risk-factors for TB and TB-HIV co-infection [8,9]. The 2016 global, regional and national burden of TB report for 1990–2016 shows a similar age distribution of incidence and mortality [10]. Screening young adults at job sites and recreational centers could potentially lead to early treatment and less impact.

The mean adherence rate of 91.4 (±5.8) % observed in this study appears to be well above the national average in Nigeria [1]. The higher rate of adherence of the present study is in contrast to a higher default rate reported in Southwestern Nigeria [11] and those reported in Europe [1]. Our improved adherence among females may be due to more attention to counseling provided by public health nurses and ability to find them at home when contacting defaulting patients as

**Table 2. Proportion of patients screened for HIV infections in the tuberculosis treatment program.**

| Year | HIV Status | | | Total n (%) |
|---|---|---|---|---|
| | HIV positive n (%) | HIV Negative n (%) | Unknown/Not evaluated n (%) | |
| 1992–2006 | 0 (0.0) | 0 (0.0) | 2388 (100.0) | 2388(100.0) |
| 2007 | 13(7.2) | 90 (49.7) | 78 (43.1) | 181 (100.0) |
| 2008 | 28 (19.9) | 105(74.5) | 8(5.7) | 141 (100.0) |
| 2009 | 24 (22.6) | 47 (44.3) | 35(33.0) | 106(100.0) |
| 2010 | 19 (18.1) | 71 (67.6) | 15 (14.3) | 105 (100.0) |
| 2011 | 13 (12.6) | 80 (77.7) | 10 (9.7) | 103 (100.0) |
| 2012 | 10(10.0) | 78 (78.0) | 12 (12.0) | 100 (100.0) |
| 2013 | 7(8.0) | 79(89.8) | 2 92.3) | 88 (100.0) |
| 2014 | 11(14.9) | 59(79.7) | 4 (5.4) | 74 (100.0) |
| 2015 | 5(5.4) | 66(71.0) | 22 (23.7) | 93 (100.0) |
| 2016 | 6 (5.7) | 99 (93.4) | 1 (0.9) | 106(100.0) |
| 2017 | 16 (16.7) | 78 (81.3) | 2 (2.1) | 96(100.0) |

Multidrug resistance TB identified via drug sensitivity test was found in only 6 patients (0.2%).

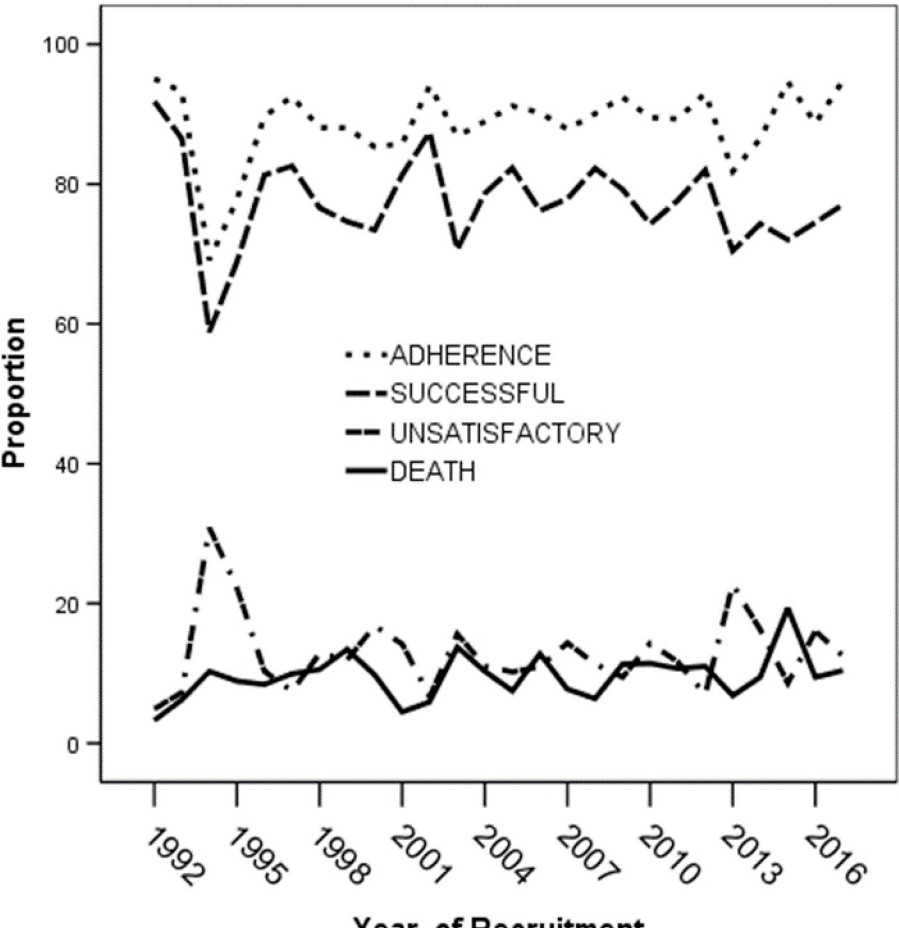

**Fig 4. Showing annual treatment outcomes for TB treatment.** Successful treatment outcome = cure+ treatment completed; Potentially Unsatisfactory outcome = default + transferred out + treatment failure.

well as their comparatively lower incidence of HIV-TB co-infection [5,6]. Data suggest health education and intensive adherence counselling are more essential for males [9].

The mean successful treatment rate observed in our study 75.3(±8.8) % is higher than the 34%, 37%, reported in other areas in Nigeria [11–14] as well as that reported in Angola and Brazil (25% and 31–71% respectively) [4]. It is similar to 76.0% mean average rate reported in Africa [15] and 71% in Europe [15]. However, it is lower than the reported success of 94%, and 86% in Bangladesh, 93% and 83% in China and 92% and 84% in Myanmar among new and previously treated cases [4]. It is noteworthy that the review period for aforementioned studies was between 1–10 years period as compared with our 25 years, which provides a better long-term perspective. The lower successful treatment outcome observed in the present study may be consequent on the inability of 10.0% of patients with treatment adherence to produce sputum as a requisite to define cure. Requesting for sputum in an asymptomatic apparently well individual may a tough and futile effort, perhaps the use of sputum induction or the development newer biomarker to define cure may be a necessity for an objective assessment of cure as a component of treatment success.

The four common predictors associated with successful treatment outcome are new pre-treatment status, female gender, EPTB and early sputum conversion similar to findings from

**Table 3. Factors associated with treatment adherence.**

| Factors | Treatment compliance | | Unadjusted | | Adjusted | |
|---|---|---|---|---|---|---|
| | Yes n(%) | No n (%) | OR (95% CI) | p-value | OR (95% CI) | p-value |
| Sex | | | | | | |
| Female | 1347(90.9) | 135(9.1) | 1.4 (1.1–1.8) | 0.002* | 1.4 (1.1–1.7) | 0.005* |
| Male | 1663(87.4) | 239(12.6) | Ref | | | |
| Age group | | | | | | |
| Children | 305(91.9) | 27(8.1) | 1.4 (0.96–2.2) | 0.076 | 1.2 (0.76–1.8) | 0.490 |
| Adult | 2705(88.6) | 347(11.4) | Ref | | | |
| TB Classification | | | | | | |
| EPTB | 474(93.7) | 32(6.3) | 2.0 (1.4–2.9) | < 0.001* | 1.8 (1.3–2.7) | 0.002* |
| PTB | 2536(88.1) | 342(11.9) | Ref | | | |
| Pre-treatment | | | | | | |
| New | 2879 (94.2) | 178 (5.8) | 5.2 (2.3–11.6) | < 0.001* | 3.1 (1.4–6.7) | 0.005* |
| Relapse | 97 (95.1) | 5 (4.9) | 6.2 (1.9–20.6) | 0.003* | 2.8 (1.1–7.5) | 0.037* |
| Transfer IN | 5 (83.3) | 1 (16.7) | 1.6 (0.16–15.8) | 0.687 | 0.6 (0.10–4.2) | 0.639 |
| Default IN | 25 (75.8) | 8 (24.2) | Ref | | | |
| Failure IN | 114 (86.4) | 18 (13.6) | 2.0 (0.79–5.2) | 0.140 | 1.2 (0.49–2.8) | 0.723 |
| Unknown IN | 49 (90.7) | 5 (9.3) | 3.1 (0.93–10.7) | 0.066 | 2.0 (0.68–5.9) | 0.203 |
| HIV Status | | | | | | |
| Positive | 144(94.7) | 8(5.3) | 1.2 (0.69–2.2) | 0.480 | 1.2 (0.65–2.1) | 0.590 |
| Negative | 793(93.1) | 59(6.9) | Ref | | | |
| Unknown | 2232 (93.8) | 148 (6.2) | 1.0 (0.78–1.3) | 0.990 | 0.9 (0.72–1.2) | 0.578 |

*statistically significant at p < 0.05, ref: Reference level. Significant variables on bivariate analysis for treatment adherence were gender, age, forms of TB, pre-treatment and HIV status were subjected to binary logistic regression with reference indicator as the male gender, adult, pulmonary tuberculosis, pre- treatment default and HIV negative respectively.

other studies [1,16]. Existing literature demonstrates prompt treatment of new cases can be highly successful with minimal treatment failures thus preventing the development of multi-drug resistant *Mycobacterium tuberculosis* [6,8,10].

The mean mortality rate of 10.0(±3.6) % in our study is higher than the reports by Olarewaju *et al* [8] and Ukwaja *et al* [17] from Nigeria. However, if the time under review (1999 and 2011) is matched with the same interval in the referenced studies [8,17], there is no significant mortality difference. The mortality peak occurred in 1993–1995 which corresponds to a period of national food crisis and intense poverty [18]. Recent reports show a clear relationship between poor sociodemographic indexes, a low gross domestic product and excess mortality in patients with TB [1,10].

The factors associated with potentially unsatisfactory outcomes, a combination of treatment default, treatment failure and transfer out in our study were adult population, TB/HIV co-infection and delayed sputum conversion were consistent with the report from Morocco by Dooley *et al* [19] and were also identified in the systematic review by Waitt *et al* [20] and others colleagues [21,22].

Despite the positive adherence rates in our study, the sub-optimal overall mean treatment outcomes call for strengthening of our TB programs, improving patient-centred care models and individualised treatment plans. It would include mandatory TB screening/diagnosis with a nucleic acid-based test (GeneXpert) and upfront drug sensitivity test with appropriate drug intervention based on standard WHO guidelines. Additionally, a comprehensive model should factor in ways of addressing structural problems such as poverty

**Table 4. Factors associated outcomes: Successful treatment outcome.**

| Factors | Successful Treatment | | Unadjusted | | Adjusted | |
|---|---|---|---|---|---|---|
| | Yes n(%) | No n (%) | OR (95% CI) | p-value | OR (95% CI) | p-value |
| Sex | | | | | | |
| Female | 1192(80.4) | 290(19.6) | 1.3 (1.1–1.5) | 0.003* | 1.3 (1.1–1.5) | 0.006* |
| Male | 1448(76.1) | 454(23.9) | Ref | | | |
| Age group | | | | | | |
| Children | 279(84.0) | 53(16.0) | 1.5 (1.1–2.1) | 0.005* | 1.3 (0.98–1.8) | 0.067 |
| Adult | 2361(77.4) | 691(22.6) | Ref | | | |
| TB Classification | | | | | | |
| EPTB | 420(83.0) | 86(17.0) | 1.4 (1.1–2.9) | 0.004* | 1.3 (1.01–1.7) | 0.036* |
| PTB | 2220(77.1) | 658(22.9) | Ref | | | |
| Pre-treatment | | | | | | |
| New | 2419(79.1) | 638(20.9) | 2.8 (1.4–5.6) | 0.004* | 2.7 (1.3–5.4) | 0.005* |
| Relapse | 81(79.4) | 21(20.6) | 2.8 (1.2–6.6) | 0.015* | 2.9 (1.3–6.8) | 0.012* |
| Transfer IN | 33(50) | 33(50.0) | 0.7 (0.13–4.2) | 0.731 | 0.8 (0.13–4.4) | 0.759 |
| Default IN | 19(57.6) | 14(42.4) | Ref | | | |
| Failure IN | 83 (62.9) | 49 (37.1) | 1.2 (0.58–2.7) | 0.575 | 1.3 (0.60–2.9) | 0.489 |
| Unknown IN | 35 (64.8) | 19(35.2) | 1.4 (0.56–3.3) | 0.500 | 1.4 (0.57–3.4) | 0.459 |
| HIV Status | | | | | | |
| Positive | 109(71.7) | 43(28.3) | 0.7 (0.51–1.1) | 0.150 | 0.7 (0.49–1.1) | 0.099 |
| Negative | 657(77.1) | 195(22.9) | Ref | | | |
| Unknown | 1874 (78.7) | 506 (21.3) | 1.1 (0.91–1.3) | 0.323 | 1.0 (0.85–1.3) | 0.730 |

*statistically significant at p < 0.05, ref: Reference level. Significant variables on bivariate analysis for successful treatment were gender, age, forms of TB, pre-treatment and HIV status were subjected to binary logistic regression with reference indicator as the male gender, adult, pulmonary tuberculosis, pre- treatment default and HIV negative respectively.

while addressing behavioural problems among high-risk patients such as HIV/TB co-infected and young male patients. It should provide a humanistic-holistic approach of psychosocial care and spiritual support already established mediators of treatment outcome in the literature [23,24]. In 2020, a high index of suspicion of MDR TB with early therapeutic drug monitoring and testing for patients with poor response is mandatory. Mandatory outreach contact tracing, annual reporting and evaluation of treatment outcomes should be evaluated as per the recommendations of the IUATLD. Annual feedback provided to the State Ministry of Health by clinicians could be used for ongoing evidence-based decisions and future planning. Finally, efforts should be made to enhance the effectiveness of existing collateral preventive interventions such as BCG vaccination, INH prophylaxis for TB contacts, and expanded screening of HIV-infected individuals, who do not yet have TB. Future efforts should also include improved funding/resources for more susceptibility test and better diagnostics/outreach in children to decrease their rates dramatically. Finally, improving overall socio-economic conditions in LMICs will ultimately be a major factor in significantly decreasing TB.

Our study limitations include missing data, lack of sensitivity test and genetic fingerprinting before commencement of treatment. Due to financial limitations, only subjects who failed treatment were subjected to sensitivity testing prior to use of Gene Xpert for multi-drug sensitivity in 2015. Additionally, a comprehensive review of HIV/TB co-infections was limited as free HIV screening only became available in 2006.

## Conclusion

Meeting SDG goals and having a world without TB requires tackling socio-economic factors including improving education, community sanitation/hygiene and reducing extreme poverty. It also requires increasing financial support to ensure adherence, equipping motivated trained personnel, providing subsidized/free TB diagnostics, and finally annual evaluation and monitoring of treatment programs throughout each region/country worldwide.

## Supporting information

**S1 Appendix. Hospital attendance and number of TB cases managed in a regional health centre.**
(DOCX)

**S2 Appendix. Factors associated outcomes: Death outcome.**
(PDF)

**S3 Appendix. Factors associated with potentially unsatisfactory treatment outcome.**
(PDF)

## Acknowledgments

We acknowledge the efforts of all the staff who have worked assiduously at the TB treatment center from the onset of the program. We also thank Professor Aderele Ilemobade Wilson, Professor Oyedeji Ademola Gabriel, Professor Cynthia Howard and Emily Danich for reviewing the manuscript.

## Author Contributions

**Conceptualization:** Michael A. Alao.

**Data curation:** Michael A. Alao, Daniel A. Gbadero.

**Formal analysis:** Yiong Huak Chan.

**Writing – original draft:** Michael A. Alao.

**Writing – review & editing:** Michael A. Alao, Stacene R. Maroushek, Yiong Huak Chan, Adanze O. Asinobi, Tina M. Slusher, Daniel A. Gbadero.

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
