## [Decision Letter · Decision Letter 0]

8 Jul 2020

PONE-D-20-11827

Treatment Outcomes of Nigerian Patients with Tuberculosis: A Retrospective 25-year Review in a Regional Medical Center

PLOS ONE

Dear Dr. Slusher,

Thank you for submitting your manuscript to PLOS ONE. After careful consideration, we feel that it has merit but does not fully meet PLOS ONE’s publication criteria as it currently stands. Therefore, we invite you to submit a revised version of the manuscript that addresses the points raised during the review process.

We look forward to receiving your revised manuscript.

Kind regards,

Tom E. Wingfield

Academic Editor

PLOS ONE

Additional Editor Comments:

Dear Tina M. Slusher and colleagues,

Thank you for your submission to PLOS One. Firstly, please let me apologise for the Covid-19 related delay in review of this manuscript. One of the reviewers had hoped to review but due to heavy clinical commitments in Nigeria had been unable to do so. I apologise for this unintentional delay.

Your manuscript is an important piece because it supports understanding of risk factors for adverse TB treatment outcome in a high TB burden setting. It is also an excellent use of programmatic data, which is fantastic to see and could be emulated by other NTPs and researchers in diverse settings.

The manuscript is well written but there are a number of mainly methodological issues which need addressing prior to re-submission for re-review. I have recommended major revisions and my comments and those of the reviewer are listed in this email below.

EDITOR COMMENTS

MAJOR

1) Table 1: is this incidence rate or number of TB cases? The right hand column says "total TB cases" suggesting the latter. I would suggest moving Table 1 to supplementary file and keeping Figure 2 which is explanatory.

I think that a more helpful Table 1 would summarise the sociodemographic (age / sex / poverty level if you have that data, smoker etc) and clinical characteristics (e.g. PTB/EPTB, HIV status, treatment category) of a) all patients b) adults and c) children. This could potentially be integrated with Table 2

2) Figure 1: unsatisfactory listed twice and numbers not clear. Don't seem to compute. Suggest splitting adult and child outcome data please.

3) Figure 2: remove decimal point and suggest incidence rate is "per 100,000 per year" as opposed to % as this will help the reader and be comparable

4) Please explain in a bit more detail how adherence was calculated for both intensive and continuation phase (e.g. TB treatment registers, if patients who were LTFU had any measure of adherence measured prior to being LTFU) and if DOTS was maintained throughout this period or whether in continuation phase there was 3x week treatment

5) Disease characteristics: I don't think that statistical calculations are necessarily needed for this comparison and don't add much to the raw numbers so would suggest dropping. If you feel strongly that these should stay in then please indicate how the differences in proportion were calculated and present the estimates of error related to these calculations.

6) Proportion of patients with HIV each year would be useful to understand, please can you add this data either in text, integrate into existing table/figure, or new figure/table

7) Table 3 is unclear. Please can the authors indicate a) what factors were adjusted for in the multivariate analysis? What is "negative" (presumably HIV)? I think tables throughout the manuscript require explanatory legends please. Please can the authors consider whether a separate analysis of the factors associated with treatment adherence and treatment success in a) adults and b) children. This could help to tailor recommendations/advice for different age groups.

8) Table 3 also highlights that the authors seem to have omitted how good adherence was calculated. It seems that they used logistic regression so it must be as a dependent binary variable so it would be useful to know what this threshold was. I think the authors should consider whether, given they have the data on adherence, a linear regression model of number of tablets taken / % adherence as the dependent variable might be more fitting. Another consideration would be a time to event (e.g. completing or stopping treatment) would be achievable with the data available.

9) I appreciate the massive efforts that this follow-up and analysis would have taken. However, given the missing data on treatment outcomes (e.g. a quick calculation of n/N for Table 3 shows there are a significant number of people without a treatment outcome). This could be dealt with in two ways: firstly, the authors could do an analysis of the sociodemographic and clinical characteristics of those with MISSING outcome data; or second, which is more straightforward, is just to mention the missing data in the limitations section and that those with missing data might represent a more vulnerable subgroup of patients.

10) All the comments above relating to Table 3 apply equally to Table 4.

MINOR

11) Referencing needs to be reviewed throughout including formatting of references in main text.

12) Please can the authors make clear in the methods if they included any interaction terms / likelihood ratio when doing their regression tables. If they did not then suggest that they do this to check for interactions between independent variables.

Journal Requirements:

2. Please provide additional details regarding partient consent. In the ethics statement in the Methods and online submission information, please ensure that you have specified (1) whether consent was informed and (2) what type you obtained (for instance, written or verbal, and if verbal, how it was documented and witnessed). If the need for consent was waived by the ethics committee, please include this information.

Reviewers' comments:

Reviewer's Responses to Questions

**Comments to the Author**

1. Is the manuscript technically sound, and do the data support the conclusions?

Reviewer #1: Yes

2. Has the statistical analysis been performed appropriately and rigorously? 

Reviewer #1: Yes

3. Have the authors made all data underlying the findings in their manuscript fully available?

Reviewer #1: Yes

4. Is the manuscript presented in an intelligible fashion and written in standard English?

Reviewer #1: Yes

5. Review Comments to the Author

Reviewer #1: The paper is generally well written and interesting to read. I recognise that is difficult to collect data from the last 25 years of treatment outcomes in Nigeria. It needs persistence and patience. The value of this work is great as provides the scientific community with useful information and conclusions on how the disease expands. The introduction is giving the reader a good, generalized background of the tuberculosis disease and the protocol and results are generally clear and well presented. However, I see the following minor issues that should be resolved before publishing this paper:

L134:”develops smear-positive TB” can be replaced with “has a positive smear”

L137: “Patient dies of any cause” can that be rephrased? Not sure how correctly expressed it is.

L173: “Testing” can be preferably replaced with “test”

L182: “Giving a M:F of 1.3:1”, can be replaced with “giving a male:female ratio of 1.3:1”

L182-183: “Children constituted 332 (9.8%)”, the meaning is not clear. Does it mean that 332 of the study population were children?

L187-188: Numbers should be expressed with words or numbers, so they match the final picture.

L199-200: Please write this sentence in a clearer way

L219-220: The word “about” should be emitted.

L249-251: Not clear meaning, not clearly written

L254: “this studies higher rate…”. Does it mean “the higher rate of adherence of these studies”?

L279: “our mean mortality rate” can be rephrased

L308: “more” can be emitted

Comment on the table with Proportion with TB-Age: Is age expressed in years? Is proportion with TB expressed in %? Please enter the units.

Comment on the flowchart: not sure/unclear the two unsatisfactory results

General comments:

Please check again the numbers of the references. They are expressed in different ways (mixed). Focus on the discussion.

6. PLOS authors have the option to publish the peer review history of their article (what does this mean?). If published, this will include your full peer review and any attached files.

Reviewer #1: No

---

## [Editor Report · Decision Letter 1]

2 Sep 2020

Treatment Outcomes of Nigerian Patients with Tuberculosis: A Retrospective 25-year Review in a Regional Medical Center

PONE-D-20-11827R1

Dear Dr. Slusher,

We’re pleased to inform you that your manuscript has been judged scientifically suitable for publication and will be formally accepted for publication once it meets all outstanding technical requirements. Thank you for addressing the reviewers' comments on your original submission.

Kind regards,

Tom E. Wingfield

Academic Editor

PLOS ONE
---

## [Editor Report · Acceptance letter]

20 Oct 2020

PONE-D-20-11827R1 

Treatment Outcomes of Nigerian Patients with Tuberculosis: A Retrospective 25-year Review in a Regional Medical Center 

Dear Dr. Slusher:

I'm pleased to inform you that your manuscript has been deemed suitable for publication in PLOS ONE. Congratulations! Your manuscript is now with our production department. 

Kind regards, 

on behalf of

Dr. Tom E. Wingfield 

Academic Editor

PLOS ONE